# A Textual Analysis for Understanding the Relations and the Identity Construction in Adolescent Oncology Patients: Retrospective Personal Views in Order to Educate Health Professionals

**DOI:** 10.3390/bs12050120

**Published:** 2022-04-21

**Authors:** Francesca Bomben, Maurizio Mascarin, Giuseppe Maria Milano, Paola Quarello, Marco Spinelli, Assunta Tornesello, Carlo Alfredo Clerici, Federico Mercolini, Domitilla Elena Secco, Maria Antonietta Annunziata, Andrea Ferrari, Marina Bertolotti

**Affiliations:** 1AYA Oncology and Pediatric Radiotherapy Unit, Centro di Riferimento Oncologico IRCCS, 33081 Aviano, Italy; mascarin@cro.it; 2Psycho Oncology Unit, Centro di Riferimento Oncologico IRCCS, 33081 Aviano, Italy; annunziata@cro.it; 3Department of Hematology/Oncology, Cell and Gene Therapy, IRCCS Bambino Gesù Children’s’ Hospital, 00165 Rome, Italy; giuseppemaria.milano@opbg.net (G.M.M.); domitillae.secco@opbg.net (D.E.S.); 4Pediatric Onco-Hematology Unit, Stem Cell Transplantation and Cellular Therapy Division, Azienda Ospedaliera-Universitaria Città della Salute e della Scienza, Ospedale Regina Margherita, 10126 Turin, Italy; paola.quarello@unito.it (P.Q.); marina.bertolotti@unito.it (M.B.); 5Pediatric Hematology-Oncology Department, “Fondazione MBBM”, Ospedale San Gerardo, 20900 Monza, Italy; mspinelli@fondazionembbm.it; 6Pediatric Oncology Unit, Ospedale Vito Fazzi, 73100 Lecce, Italy; assuntatornesello@gmail.com; 7Department of Oncology and Hemato-Oncology, University of Milan, 20133 Milan, Italy; carloalfredo.clerici@istitutotumori.mi.it; 8Pediatric Oncology Unit, Fondazione IRCCS Istituto Nazionale dei Tumori, 20133 Milan, Italy; andrea.ferrari@istitutotumori.mi.it; 9Department of Pediatric, Ospedale Centrale di Bolzano, 39100 Bolzano, Italy; federico.mercolini@libero.it

**Keywords:** adolescents, AYA oncology, textual analysis, experience of disease, professional education

## Abstract

Patient input is critical for all aspects of value-based healthcare design. This contribution describes the following: the specifics of communications with doctors regarding the disease in adolescents and young adults with cancer; the patients’ thoughts, emotions and changes in self-perception; “other meanings” taking shape along the treatment pathway; and reacting modes to the disease and treatments. Thirty-five Italian AYA patients in follow-up (age 18–24) were involved in a plenary interview on the cited aspects of their oncological experience. The answers were analyzed by MADIT (Analysis Methodology of Computerized Textual Data) with the software SPAD. MADIT allowed us to perform text analysis, describe the graphical outcomes and discuss the results. Respondents took a first-person perspective and their personal narrative recall had objective and unequivocal connotations. Experience was narrated mainly by maintenance repertoires that fix the reality of disease, its treatments and personal identity. The account focused on the tumor and on an agreed approach to it. The time “after” was described as a distressing space that defines them. Making sense of the events was considered a significant help. Professionals need to focus on the discursive repertoires of communication with which the inner and outer reality are built. Lastly, these patients required a two-way dialogue throughout the entire caring process.

## 1. Introduction

Every year in Italy, there are approximately 2,857,000 adolescents and 2,957,000 young adults. Among these, there are approximately 800 and 1100 new cases of cancer between the ages of 20 and 24 and 15 and 19 each year, respectively [1,2]. The main diseases that occur in these age groups are: leukemia, lymphoma and reticuloendothelial neoplasm, central nervous system tumors, malignant bone tumors soft-tissue and other extra osseous sarcomas, germ cell, trophoblastic and other gonadal neoplasms and other malignant epithelial neoplasms and melanoma (AIRTUM database 2013–2017). Adolescents with cancer form a subgroup of patients whose clinical management and access to the best possible treatment remain a challenge, especially when compared with improvements related to the children’s care model achieved over the years [3,4,5]. They face complex, specific medical and psychosocial challenges due partly to the nature of this developmental stage in life, and partly to the lack of improvement in the survival rates for these age groups, which differentiate them from pediatric and older adult cancer patients. In Western countries, since the turn of this century, there has been a more substantial effort to research and develop dedicated programs for adolescents and young adults (AYA) with cancer, engaging with healthcare operators, stakeholders and the patients themselves [3,5,6,7,8,9,10]. In Italy, based on the first experience of the Youth Area Project at the Centro di Riferimento Oncologico in Aviano [10] (developed within an adult medical oncology setting) and the Youth Project [7] at the Istituto Nazionale Tumori in Milan (established at the pediatric oncology unit), other cancer centers have introduced projects specifically tailored to adolescent patients, where a psycho-educational plan can improve the global health of the patients, families and professional caregivers. When AYA are newly diagnosed with cancer, the patients are taken into care at an appropriate treatment center with the aims of eradicating, or at least containing, their disease as much as possible, limiting its sequelae, adjusting the patient’s developmental process, and providing the clinical support needed to restore the patient’s overall health. The challenges of care take on particular features when the patients involved are adolescents, demanding even more thorough and expert attention to the individual’s interactive, systemic and relational processes [8,9]. Devising an active role for patients in their treatment pathways can facilitate their compliance, adaptability and psychological well-being, with benefits extending to various aspects of their lives. This intent is achieved through a value-based health design: patients are the protagonists of the path of illness and treatment, and for this reason they are the main interlocutors of any reflection on the needs or criticalities of their own experience [9].

Given the above-mentioned experimental aims, this contribution is based on previously published reflections and research [4,5,6,11,12,13], and aims to deepen some specific issues through a qualitative-quantitative way that gives patients the role as protagonists of the research itself. The results of this investigation allow a reflection on the construction of personal identity, especially in adolescence. In the literature, this is a much-discussed issue, which is accessed from different theoretical positions. The present research starts from the assumptions of post-modern psychology [14,15,16,17] and is based on a constructionist paradigm [18,19,20], according to which personal identity is constructed in a dialogical sense, within relational and hermeneutic contexts [19,20,21]. These reflections have moved researchers: what teenagers experience, in their own experience of illness, can become a characteristic and constitutive element of their own idea of themselves and, consequently, of their own personal identity. Knowing the characteristics of their narrations allows healthcare professionals to act in a favorable and functional way also for the psychological well-being of the patient.

## 2. Materials and Methods

In this study, the researchers describe four macro-areas relating to this particular oncological setting, covering topics emerging from the literature as important and characteristic of the age group of our respondents [3,5,8,12]. These macro-areas concern: (1) the specifics of communications regarding the disease between patients and doctors; (2) patients’ thoughts, fears, and changes in their self-perception; (3) “other meanings” taking shape along the treatment pathway; (4) modalities for reacting to the disease and its treatment. Therefore, this study aims to describe how young cancer patients relate their experience of illness and treatment starting from the specific inherent topics offered to them. The consequent strategy of the researchers, in short, was to set up a focus group, implemented through a semi-structured interview. The national medical conference “Adolescents and Cancer”—held in Rome in November 2019—was the experimental site of the present study, which aimed to examine how adolescent cancer patients in follow-up describe their experience of disease, as we have already discussed in a previous work [11]. Each Italian AIEOP center (AIEOP is the Italian acronym of *Associazione Italiana Emato-Oncologia Pediatrica*—Italian Association of Pediatric Hemato-Oncology) notified its patients of the interview initiative and gave each of them free choice to participate. The patients contacted were aged between 18 and 24 years and in treatment or in follow-up for cancer. The patients defined as “in treatment” had just completed this phase of the antineoplastic process, and for this reason they were included in the experimental group: their experience of active therapy was already over. The adhesion that occurred was the method of constructing this convenience sample: the group of participants were cancer patients who voluntarily agreed to participate in the interview. No further socio-demographic data for the participants were recorded, as they were not necessary for the purposes of the research, its aims and its methodology. The criteria for inclusion were as follows: age (18–24), experience of oncological disease (final phase of treatment and follow up), execution of treatment at an AIEOP center, voluntary participation in the interview and in the research that would ensue. Indeed, the specific nature of our investigation was explained before the interview, and they were told that the interview would be audio recorded. Each participant signed their informed consent. According to institutional rules, the study was approved by the ethics committee (ID 3872, CRO 21–39). The proposed ad hoc questionnaire contained 14 questions covering the four macro-areas identified on the strength of the relevant literature [4,12,18,22,23] (Table 1). A conference session was dedicated to the interview: each question was allotted up to three minutes, and participants were free to raise their hands and speak whenever they wished, one by one. When the matter posed by the first question no longer produced other requests for intervention, the second question was asked, and so forth. No one was forced to answer any questions and the interventions were all voluntary. A physician (on the staff of one of the AIEOP centers involved) posed the questions one at a time, projecting them on a screen in the conference room. Where necessary, another physician specializing in clinical psychology and psychotherapy would add some comments to deal with any emotional and narrative “fallout” from the respondents’ stories. The whole session was subsequently transcribed to enable a detailed analysis of the text content in relation to the study aims.

The MADIT application (Methodology of Analysis of Computerized Textual Data) [22], within the theoretical-epistemological frame of the dialogical model [21,22], was used to design the study and as the tool for collecting data, and to complete the whole process (text analysis, description and discussion of the results). The size of the sample depended on it, and for MADIT, a minimum number is not contemplated in correspondence with the theoretical and practical aims of the research methodology adopted [18,23]. MADIT is a scientifically based method [18,21] equipped with precise criteria that allows the analysis of textual data to achieve the objectives set by the research. This methodology analyzes discursive productions on two levels: a content level (what is said) and a procedural level (how that content is expressed). This second level of analysis makes it possible to measure whether the considerations offered by the respondents are static, i.e., whether they tend to remain unchanged over time or they are susceptible to change.

We first obtained a specific vocabulary for each of the four macro-areas of interest. Then, we separately conducted a lexical correspondence analysis on each vocabulary, using a quali-quantitative method to elucidate the textual data. The resulting graphs and tables identified and described the clusters of meaning and discursive repertoires that emerged as characteristic of our participants’ answers [18,21,22,23]. The former concern the thematic/explanatory content voiced by participants, the latter the narrative modalities they used to do so, which reveal how much they affected the individual’s identity and health. The textual analysis, by content and process, was done with the informatical support of SPAD software (Systeme Portable Analyse Donnèe [24]—Textuelle), a computer tool for the lexicometric analysis of written texts. The reported results represent what emerged from the analysis of the 55 answers the young patients gave. The graphs generated by the software stem from identifying the factors that—based on the statistical calculations performed by the software—give us an outline of the whole architecture of the texts analyzed [18,23]. Each of the four macro-areas considered thus produced a distinct graphic (Figure 1 and Figure 2), thereby satisfying one of the specific goals of our study. The highly significant linguistic forms in each graph are projected in the areas furthest away from the point where the two axes cross over because the outliers (of very weak significance) had already been deleted before starting the correspondence analysis to avoid a bias in the end result [18,22,23].

## 3. Results

The group of respondents comprised 35 cancer patients that voluntarily joined the survey. From the moment each question was posed, all the questions elicited a manifest willingness of the participants to express themselves in the first person. Across topics, each answer began with a more or less detailed account of the patient’s cancer experience, including important moments, multiple individuals, and simple and complex emotions. The figures presented are the output generated by the software that the researcher reads through MADIT. The categories set as a reference for the analysis process are red, and the text excerpts (in synthetic labels) are blue. To read these results with significance, it is necessary to observe the position of the projected excerpts, considering that the distance from the origin of the cartesian axes is a measure of the significance of the excerpt itself in the general architecture of the answers. In the first macro-area, i.e., “the specifics of communications regarding the disease between patients and doctors”, it emerged that communication processes regarding the disease adopted a reporting style (“I was”, “I did”, “they said”, “I felt”, etc.). Respondents used a first-person perspective and unchanging terms, as if their personal narrative recall had objective and unequivocal connotations. When speaking about when their cancer was first diagnosed, they mentioned the disease by name, their emotions, the presence of their parents, the verbal and proxemic specifics of the dialogue, their friends, and—even more significantly—what was happening to them, and their prompt and autonomous recourse to searching the Internet. This moment was closely connected to when they first shared the news with others (sometimes giving their reasons “why”) and to the feeling that they had to do something “immediately” to manage and announce it at school. The full realization of the severity of their disease occurred during the period when they developed symptoms. It was conveyed through their parents and reified in their encounter with the medical world, their treatments, and the hospital ward. It emerged with more feeling in the hopes they experienced at the end of their treatment and during their follow-up. The explanations regarding the time “after” (as opposed to “before”), once their active treatment came to an end, are incomplete—a space in which patients define themselves more precisely, revealing the perception that they see this phase as a consequence of their treatment.

The patients’ thoughts, emotions and perceived changes were also narrated mainly by means of maintenance repertoires that fix the reality of their personal identity, their experience and its interpretation (the topic of the second macro-area, “patients’ thoughts, fears, and changes in their self-perception”, Figure 2). The cognitive effort contained in the memories they described focused on coping with the situation, with massive use of impersonal and transversally attributed verb forms (“you have”, “they tell you”, etc.), especially around the patients’ fears and their treatments. Nevertheless, there are significant signs of opportunities opening up, which are echoed in phrases about “outside” and beyond the disease. The emotional sphere is interwoven with the patients’ thoughts when naming their experiences by using terms such as “blackout”, “cold shower”, or “total darkness”, and in this discourse they included the figure of their physician and the city where their treatment center was located (see the linguistic label “Roma”). Moving towards the changes the patients perceived in themselves, they continued to adopt a reporting style that described their situation in absolute terms (“everything”, “only”, “so”, etc.). They mentioned changes in their character (expressed through virtues such as strength, courage, etc.) and changes in their body, which are far more significant, since they felt they had to face them largely alone.

In the third macro-area, “other meanings taking shape along the treatment pathway”, (Figure 3) (the role of other people in being helpful, serving as a reference figure, or accompanying the adolescent patients on their treatment pathway is also described as something fixed and unquestionable. In the case of the people that helped, the patients’ accounts focused on the disease and on an agreed approach to it. Their words hinge on luck, life, friends, classifying and charting the whole in virtual interactions and daily presences. References to healthcare providers included numerous variables: doctors, nurses, psychologists, residents and educators, validated for sharing their time, spaces, and suggestions on how to manage treatment side-effects. This aspect is interwoven with contact with other patients who were especially exposed to the patients’ questions, given their experience and proximity in the hospital. In the respondents’ words, their friends were involved in the discussion of what will come “after”, in diverting activities during the treatment (see linguistic label “cinema”, which was also indicative of many other similar activities) and when talking about their “fears” for the future.

Finally, regarding the topic of the fourth macro-area, “modalities for reacting to the disease and its treatment”, the following was observed. The reaction modalities explored in our investigation (classified under the labels “to swear”, “to write” and “to execute”) emerged in categorical schematics that refer actions to the present (Figure 4). Regarding the first label, patients made significant mention of the often-unwanted advice they received from people outside and the important bonds that were built. They also hinted at certain moral obligations—particularly revolving around their use of the expression “shut up” applied to mothers, among others—that the patients had linked in some way to their hypothetical fault for what had happened to them. Writing emerged as a coping mechanism that prompted many respondents, after completing their treatment, to redefine and/or justify themselves, identity determinations of the moment (“mad”, “adolescent”, etc.), and in adjectives describing what had happened to them (see “important”). Writing was seen as an expedient for making sense of the events, the experience and the crisis, and the results of our analysis suggest that both the topics and the writing format used are significant. On non-compliance with the recommendations of health operators, the respondents used verbs in the past tense and generalized discursive forms, such as “always” and “anyway”. In this area, there was an emphasis on transgressions regarding food, symptoms, school and therapy management.

## 4. Discussion

A cancer diagnosis is a critical event that threatens an individual and interferes with all the domains around which their life revolves: physical, psycho-emotional and relational [4,25,26]. Recently, the European Commission of the EU launched the childhood cancers and cancers in adolescents and young adults initiative to increase understanding of pediatric and AYA cancer [27,28].

This contribution has observed the weight of some possible complementary care actions to the medical aspect [3,4,5,12,28,29] that, specifically, healthcare staff wanted to investigate and understand: (1) care of communication between doctor and patient is an access key that lays the foundation for a functional relationship between the parties that is useful for compliance and adherence to treatments [3,8,9,10,11]; (2) attention to thoughts, fears and changes in self-perception guarantees a greater understanding of the impact of the side effects of the medical procedures, as well as of the behavioral reactions of the patients [3,9,10,12,25]; (3) adequate consideration of the presence of other individuals involved in the treatment process can help the staff manage the simple or complicated sequelae of treatments, and favors the continuation of normality for each patient, offering a positive psychological impact [3,6,9,10,11,12,25]; (4) subjective ways in which each patient copes with the disease and its treatments allows each operator to understand and refrain from judging these actions, both in the present and in the planning sense [3,4,5,7,9,10,25,26,29].

This study allowed us to understand the point of view of young patients on topics that the literature [7,8,9,10,11,12] defines as central to adolescent oncology (Table 1); thus, we can intercept not only the relevance, but also the perception of the immutability/changeability attributed to each aspect. Based on the results obtained, a multidisciplinary approach to patients becomes a must, allowing the team to deal with health and not just the disease of the body [8,9,12,25]. Different and complementary views allow taking charge on several levels, which are connected and interdependent with each other. This multidimensional care is the basis of compliance, adhesion and adherence to treatments, and becomes a predictive factor for the well-being of the “after”. The interaction with healthcare staff on multidisciplinary topics allows young patients to continue the process of building their own identity by inserting their experience of the disease in their autobiography in a coherent and constructive way.

Cancer care and education are no longer the responsibility of the health sector alone. It requires engagement and buy-in from a wide range of sectors and stakeholders. It should be inclusive, giving everyone concerned a stake in its success and, above all, it should be people-centered, focusing on the patient’s needs throughout [18]. Furthermore, the US National Academies’ reports emphasize that practices for adolescents should create incentives for discovery and innovation. Its review, “Promoting Positive Adolescent Health Behaviors and Outcomes”, found that programs teaching social and emotional skills—including self-awareness, self-management, social awareness, relationship skills, and responsible decision-making—may be particularly successful in promoting positive youth development [5,9,10,13,28,29].

In order to describe how young cancer patients relate their experience of illness and treatment, we started with a group of specific inherent topics, and our findings illustrate how the reality of cancer declined in its communicational, relational and identity-related aspects. Their responses to our questions described this as something that developed from the moment they were informed about their diagnosis, but it did not stop there [7]. It involved different individuals whose roles were perceived as important: parents, friends (who quantitatively occupied the most space in the patients’ discourses) [3,11,21,29], healthcare personnel, and other patients. Memories of what happened to them permeated all narration. It was seen as an experience that was not amenable to future review but strictly reified in the here and now of the interview (when the patients were in follow-up). A demand for personal autonomy, especially in the cognitions, actions and the management of their emotional state, was described as necessary and actively sought—which is to be expected given the characteristics of the developmental age of our respondents [9]. Friendships were classified in two ways, depending on the ability attributed to the individual concerned for understanding a patient’s circumstances and/or to face it. The cancer treatment pathway was described as starting not so much from how effortful it was, but rather, from what the patients had been obliged to give up and the changes the treatments had imposed. As the young people themselves explained, this is what prompted both their unspoken, circumstantial acts of non-compliance and their subsequent rethinking, which led them to justify the orders of doctors they had received in the past. Consistent with this picture, they described what they heard from other patients as information they had sought and accepted more willingly on the strength of its greater tangible reliability. The transcription of the patients’ experiences in written stories and notes were also described as an opportunity for self-cognition and patient education.

The use of “definitive” expressions is generative of immutable perceptions, and this has connections in the text with reports about the discomfort and difficulty in reconstructing one’s daily life in the future. In this way, the personal identity of the respondents was also connoted, since it was often built on the univocal basis of the experience of illness. This experience demonstrates that patients can educate their professional caregivers and narrations in the first person about diagnosis, help, needs, communication and relationships must become the focal point of care.

Our complete investigation lacked confidentiality features, which represents a limitation of the study itself. Nevertheless, we chose to proceed despite this, as this design allowed the young patients involved to express themselves in front of many of the individuals they mentioned in the relational dynamics they described. Furthermore, this enabled them to describe experiences that are usually only discussed in a more exclusive manner with a single professional, or not at all. In future investigations, we propose to further this research, focusing on what patients configure and define as “after” because, in the present study, this phase unexpectedly emerged transversally as being more complex, more left to chance and the great unknown. One proposal for an in-depth study of this theme is the development of a subsequent investigation that compares the texts of the patients (during treatments) with those of the survivors (in follow-up) in two distinct groups. In order to intercept any specific discursive modalities, the sense and significance attributed to the “after” should be faced in an even more peculiar way.

## Figures and Tables

**Figure 1 behavsci-12-00120-f001:**
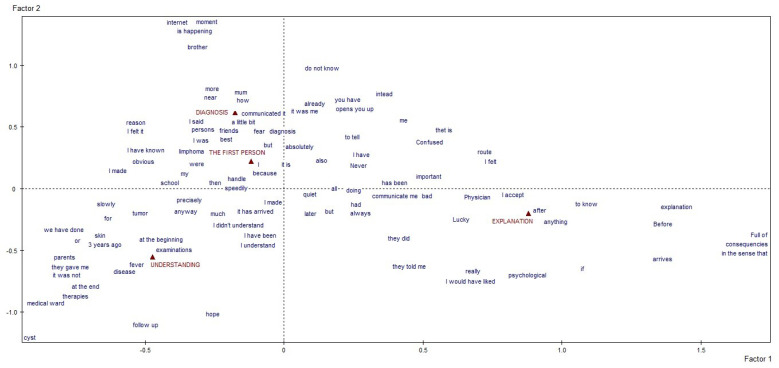
Graphic representation of the first outcome, about “the specifics of communications regarding the disease between patients and doctors”.

**Figure 2 behavsci-12-00120-f002:**
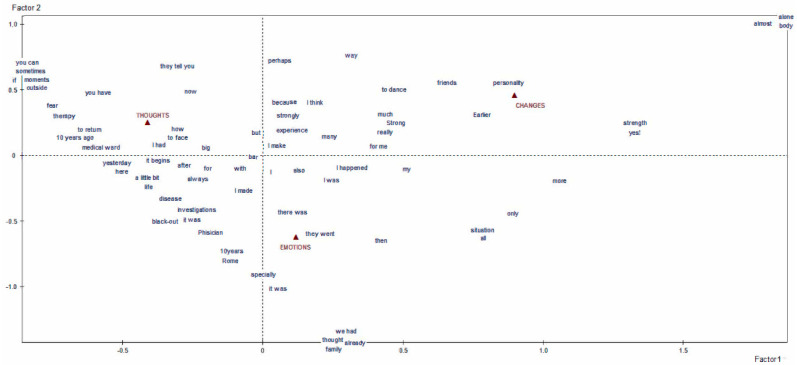
Graphic representation of the second outcome, about “patients’ thoughts, fears, and changes in their self-perception”.

**Figure 3 behavsci-12-00120-f003:**
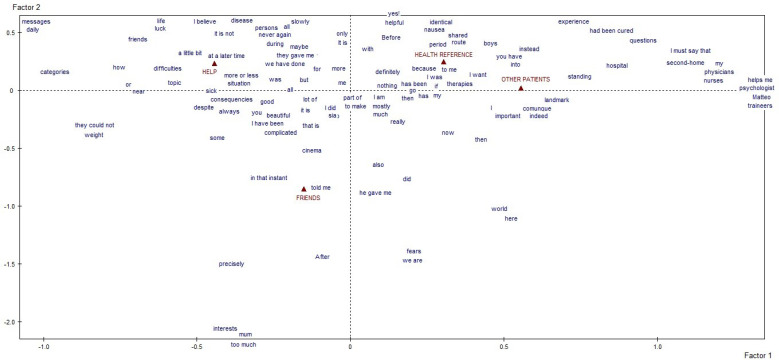
Graphic representation of the third outcome, about “other meanings taking shape along the treatment pathway”.

**Figure 4 behavsci-12-00120-f004:**
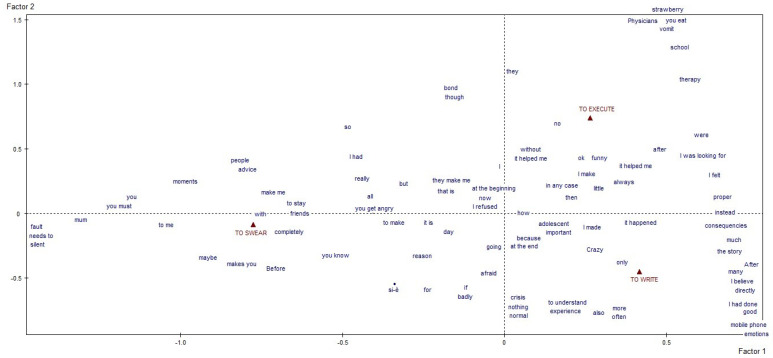
Graphic representation of the forty outcome, about “the modalities for reacting to the disease and its treatment”.

**Table 1 behavsci-12-00120-t001:** Questions posed to patients in the interview, covering the objectives of the four macro areas.

N°	Area Objectives	Questions
1	To identify the specifics of patient-doctor communications along the cancer treatment pathways of adolescent patients in follow-up (communications between healthcare operators and patients concerning topics relating to the disease and its specifics)	When and how did you find out you had a severe disease? How were you told about your diagnosis, and who was with you? Who was the first person you told about your diagnosis, apart from your parents, and how did you do so? Were you missing some important explanation? If so, what about? Was there a question that you didn’t have the courage to ask?
2	To identify the thoughts, fears and feelings about themselves of adolescent cancer patients in follow-up relating to certain stages of their treatment pathway (thoughts, emotions and self-perceptions)	What was your first thought? Was there a time when you were afraid? How has your self-perception changed, physically and in your character? How did you notice the change?
3	To identify the role of significant others for adolescent cancer patients in follow-up relating to their treatment pathway (others: family, friends, healthcare personnel, other patients-impact, needs, requests)	What about your friends? How did they react to it all? What have you found helpful? Who, among the healthcare providers, served as your reference figure during your treatment? Would you have liked to speak to someone who had been through the same experience as you?
4	To identify some specifics concerning the active role of adolescent cancer patients relating to their treatment pathway (their willingness to take action; the patient’s active/passive role when faced with their situation)	Have you ever cursed about what was happening to you? If so, who or what were you cursing? Have you ever written anything about what has happened to you? As adolescents, we all do things we shouldn’t. Have you ever done anything the doctors told you not to do?

## Data Availability

The interview has been audio-recorded and stored in Centro di Riferimento Oncologico Aviano, Italy, working site for the first and corresponding author. Code availability: SPAD Systeme portable pour l’analyse des donnees (SPAD v 5.6).

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
