# Peer review of "A Textual Analysis for Understanding the Relations and the Identity Construction in Adolescent Oncology Patients: Retrospective Personal Views in Order to Educate Health Professionals"

_behavsci, 2022, doi:10.3390/bs12050120_

Round 1

Reviewer 1 Report

I appreciate the topic of the submitted paper and find the understanding of relations in identity construction among oncological patients substantial. I also consider the research method used to be adequate to the problem under study. 

To improve the quality and impact of the paper I have several recommendations:

1.) In the abstract the value-based healthcare design is mentioned. This approach is not mentioned in the introduction. It is obvious the values and value-based approach stands in the ground of the topic it would be useful to add it into the introduction as a part of the theoretical background.

2.) The paper deals much with a personal identity construction which is not mentioned in the introduction although the discussion in contemporary psychology is wide and inspiring. 

3.) The discussion seems to be quite poor in several parts. I recommend a much deeper analysis of impacts which should be connected better to contemporary discourse in the field (e. g. the part of discussion on lines 220 - 232 which is more presentation of the results than discussion).
4.) There are some textual and structural mistakes in the text, f. e. part 5 Patents which is empty (?)

Author Response

Dear Rewiewer,

Thank you so much for your comments. We appreciate it, in particular for the peculiar attention to each aspect of the research and its methodology.

MADIT is a new methodology in this field, and for this reason we understand the possible difficulties in accessing the specificity of MADIT steps. By implementing the required revisions, we hope to have increased the readability and comprehensibility of our contribution.

We accepted the suggestions and made the consequent changes to the text. Also, we  corrected any misprint or mistakes mentioned.

I appreciate the topic of the submitted paper and find the understanding of relations in identity construction among oncological patients substantial. I also consider the research method used to be adequate to the problem under study. 

To improve the quality and impact of the paper I have several recommendations:

  1. In the abstract the value-based healthcare design is mentioned. This approach is not mentioned in the introduction. It is obvious the values and value-based approach stands in the ground of the topic it would be useful to add it into the introduction as a part of the theoretical background.

This approach is now mentioned also in the introduction, lines 70-73.

  1. The paper deals much with a personal identity construction which is not mentioned in the introduction although the discussion in contemporary psychology is wide and inspiring. 

This identity construction theory is now cited (lines 87-97), with the corresponding references in the bibliography (14,22,23,24,25,26,27).

  1. The discussion seems to be quite poor in several parts. I recommend a much deeper analysis of impacts which should be connected better to contemporary discourse in the field (e. g. the part of discussion on lines 220 - 232 which is more presentation of the results than discussion).

We agree with the reviewer’s comment. We have enhanced the discussion as suggested,  lines 264-272.

  1. There are some textual and structural mistakes in the text, f. e. part 5 Patents which is empty (?).

Thank you, we have fixed these mistakes.

Best regards,

FB

Reviewer 2 Report

Review of Research Article

“A textual analysis for understanding the relations and the identity construction in adolescent oncology patients: retrospective personal views in order to educate health professionals” 

During the analysis of the materials of this study, presented in the article “A textual analysis for understanding the relations and the identity construction in adolescent oncology patients: retrospective personal views in order to educate health professionals”, I come to the conclusion that the article title and abstract are appropriate.

The purpose of the article and its significance is stated clearly. The study methods are sound and appropriate. The writing is clear and concise. The conclusions are accurate and supported by the content. The article is of interest to members of the education research community. The study was approved by the Ethic Committee (ID 3872, CRO 21-39).  

I recommend Research Article “A textual analysis for understanding the relations and the identity construction in adolescent oncology patients: retrospective personal views in order to educate health professionals” for publication on the pages of the International Journal of Behavioral Sciences.

Author Response

Dear Rewiewer,

Thank you so much for your comments. We appreciate it, in particular for the peculiar attention to each aspect of the research and its methodology.

MADIT is a new methodology in this field, and for this reason we understand the possible difficulties in accessing the specificity of MADIT steps. By implementing the required revisions, we hope to have increased the readability and comprehensibility of our contribution.

Best regards,

FB

Reviewer 3 Report

The manuscript shows a survey focused on AYA diagnosed by cancer. It is an interesting topic, and its treatment and protocols implementation are needed. However, I have some suggestions:

Introduction

Delete full stop before the reference.

(AIRTUM database 2013-2017) must be written in square brackets.

Line 49, reference 21? If it is a mistake, please correct it.

There are several phrases without their reference, please add them (line 49-53 for example).

Line 68: take care about writing, there is no space before the reference

Material and methods:

AIEOP: explain it the first time you use it.

Could be a bias the difference between patients in treatment and patients in follow-ups? Because this clinical situation is not the same for them.

There is no inclusion or exclusion criteria.

“The adhesions that occurred were the method of constructing the sample: the 91 group of participants were 35 cancer patients who voluntarily agreed to participate in 92 the interview” that correspond to results section.

Why 35? Is there any sample size calculation?

Procedure is not clear. Do all participants answer the survey together? Is the survey individualized? Are the questions mandatory? Could be any question without answer? If there is a group session, is there any debate?

How was data analysis?

Results

There is no sample explanation, age, number, demography data….

Maybe macro areas should be explained better in methodology section in order to a better understanding of the results.

A table as a summary could help.

Discussion

There are no spaces before references.

Discussion must be supported by the literature.

“Our complete) investigation”, “Nevertheless, he researchers “ revise writing.

There is no conclusions related to the aim of the study.

References

References does not follow the journal guidelines.

More literature cites should be added to support this investigation.

Thank you for your work.

Author Response

Dear Rewiewer,

Thank you so much for your comments. We appreciate it, in particular for the peculiar attention to each aspect of the research and its methodology.

MADIT is a new methodology in this field, and for this reason we understand the possible difficulties in accessing the specificity of MADIT steps. By implementing the required revisions, we hope to have increased the readability and comprehensibility of our contribution.

We accepted the suggestions and made the consequent changes to the text. Also, we  corrected any misprint or mistakes mentioned.

Introduction

Delete full stop before the reference. Ok, we did.

(AIRTUM database 2013-2017) must be written in square brackets. Ok, we did.

Line 49, reference 21? If it is a mistake, please correct it. The exact reference is not 21 but 12, and 5. It is now correct.

There are several phrases without their reference, please add them (line 49-53 for example). Ok, we did.

Line 68: take care about writing, there is no space before the reference. Ok, we did.

Material and methods:

AIEOP: explain it the first time you use it. Ok, we did.

Could be a bias the difference between patients in treatment and patients in follow-ups? Because this clinical situation is not the same for them. This point is explained in lines 105-109. This differentiation is also mentioned in the "discussion" paragraph (lines 375-380), with the proposal of a further study with the aim of detecting  differences in any texts produced by patients in treatment and patients in 6 months and over follow-up.

There is no inclusion or exclusion criteria. The clarification of inclusion and exclusion criteria is now presented in lines 111-113.

“The adhesions that occurred were the method of constructing the sample: the group of participants were 35 cancer patients who voluntarily agreed to participate in the interview” that correspond to results section. We thank the reviewer for this suggestion: in the “materials and methods” paragraph, we describe the construction of sample (that is a methodological step of MADIT), and in the “results” paragraph we report the size of the sample.

Why 35? Is there any sample size calculation? This point is explained on lines 161, and 134-136. It is a convenience sample and its calculation is not made before but after, according to MADIT criteria.

Procedure is not clear. Do all participants answer the survey together? Is the survey individualized? Are the questions mandatory? Could be any question without answer? If there is a group session, is there any debate? This point is now explained on lines 169-178.

How was data analysis? This point is explained on lines 143-161, according to MADIT methodology and its application with SPAD software.

Results

There is no sample explanation, age, number, demography data…. This point is explained in “materials and methods”, lines 111-115, where the methodologically founded  choices are explained.  

Maybe macro areas should be explained better in methodology section in order to a better understanding of the results. We thank the reviewer for this suggestions: macro areas were described in “introduction”. Now, we have reported macro areas explain in "materials and methods". In the "results" paragraph, the areas have been recalled again to allow the reader to follow the draft.

A table as a summary could help. Ok, we did.

Discussion

There are no spaces before references. Ok, we did.

Discussion must be supported by the literature. We thank the reviewer for this suggestion: now we report citations, point by point.

“Our complete) investigation”, “Nevertheless, he researchers “ revise writing. Ok, we did.

There is no conclusions related to the aim of the study. The clarification of this point is now shown in lines 261-272.

References

References does not follow the journal guidelines. Ok, we did.

More literature cites should be added to support this investigation. Ok, we did.

Thank you for your work. Thank you for these clarifications required. The suggestions helped a lot.

Best regards,

FB

Round 2

Reviewer 3 Report

Dear authors, 

Thank you very much for taking my suggestions into account. I have a better understanding of your manuscript.

Good job!